# Use of Health Services and Rehabilitation before and after the Beginning of Long-Term Sickness Absence—Comparing the Use by Employment and Disability Pension Transition after the Sickness Absence in Finland

**DOI:** 10.3390/ijerph19094990

**Published:** 2022-04-20

**Authors:** Riku Perhoniemi, Jenni Blomgren

**Affiliations:** The Social Insurance Institution of Finland, 00250 Helsinki, Finland; jenni.blomgren@kela.fi

**Keywords:** long-term sickness absence, health care, rehabilitation, employment, disability pension, labour market status, socioeconomic determinants, trajectories, longitudinal

## Abstract

The objective of the study was to follow the health care and rehabilitation use before, during and after long-term sickness absence (LTSA), and to compare the use by post-LTSA labour market situation in terms of disability pension and employment. Individuals aged 18–58 with a ≥30-day LTSA spell in 2015 (*N* = 2427) were included from the total population of the city of Oulu, Finland. Register data included LTSA spells, outpatient health care visits, inpatient care spells and rehabilitation spells, disability pensions (DP), employment dates, and demographic, socioeconomic and disability-related covariates. The study population was followed for one year before, and three years after the start of LTSA. Negative binomial regression models were utilized to examine covariate-adjusted use of the three service types and group differences. The use of outpatient health care peaked at the start of the LTSA spell, and adjusted for covariates, the height of the peak was similar regardless of post-LTSA labour market situation. Adjusted for covariates, those who transferred to permanent DP after an LTSA used more outpatient (predicted mean 4.87 for attendance days quarterly, 95% CI 4.36–5.38) and inpatient (predicted mean 84 days quarterly, 95% CI 0.62–1.06) health care than others during three years after the start of LTSA. Individuals not employed after an LTSA showed the highest and increasing level of rehabilitation use. The results indicate that Individuals returning to employment after an LTSA are provided with relatively high amount of early outpatient care, possibly aiding the return. For individuals not employed after an LTSA, rehabilitation is used quite frequently but rather late in the disability process. The frequent use of health care among future disability pensioners is consistent with their increasing health problems leading to retirement.

## 1. Introduction

In OECD countries, disability benefits, health care and rehabilitation services cause great public expenses [1,2]. Long sickness absence spells and a high frequency of health care use are also indicators of a risk for permanent disability [3,4,5,6,7] and societal expenses. Thus, in the literature, research shows that sickness absences and a higher frequency of health care use have often been studied as risk outcomes themselves [8,9,10,11,12,13,14,15,16,17]. The association between long-term sickness absence (LTSA) and health care use has been examined as well, both internationally and in a Finnish context. More frequent health care use is associated with more frequent and longer sickness absence spells later [7,12,18,19,20]. Vice versa, longer sick leaves and disability pensions (DP) are associated with increased health care use [14,17,21,22]. However, studies are scarce concerning the temporal associations between LTSA and health care use.

Similarly, studies on the temporal associations between LTSA and rehabilitation actions are lacking, although early vocational or medical rehabilitation are highlighted in Finnish policies. In the Nordic countries, multidisciplinary and vocational rehabilitation have shown modest effectiveness on return to work, inter alia [23,24], and that rehabilitation in the early stages of sickness absence may be important for recovery [25]. In Finland, studies on the timing of rehabilitation in Finland have shown that the amount of early vocational rehabilitation can be insufficient both for future disability pensioners [26,27] and rejected DP applicants [28,29]. 

More longitudinal studies are needed in order to know how disability benefits, health care and rehabilitation services together succeed in supporting work ability and resumption of employment after disability. It is not understood how health care use develops before and during an LTSA spell or how rehabilitation is timed in relation to LTSA, nor is it well known how the use of health care or rehabilitation varies according to the labour market outcome of the LTSA spell. Do those who return to employment, those who do not and those who end up on disability pension have distinctive health care or rehabilitation use trajectories? Is the labour market outcome of the LTSA spell associated with health care use or rehabilitation use levels when the roles of LTSA duration, LTSA diagnosis, or demographic, socioeconomic and disease-related covariates [8,9,10,11,12,13,14,15,16,17,30,31,32] are examined as well? Following health care and rehabilitation use before occupational disability can reveal distinctive profiles for groups with differing labour market outcomes of sickness absence. It may also identify groups in risk of permanent disability or marginalization (i.e., those with lowered work ability outside employment). 

Only a few studies have examined changes in health care use in relation to disability benefits. Perhoniemi and Blomgren [30] showed a high level of outpatient health care use both before and during LTSA for individuals with a statutory maximum length of LTSA. The use of health care has been also shown to decrease but remain high after a disability pension transition [33,34,35]. Still, register-based follow-up studies examining how the use of health services and rehabilitation develop before LTSA and as LTSA progresses in particular are still lacking.

When examining the level of health care and rehabilitation use in relation to LTSA, it is necessary to account not only for the chronic diseases of the individual and the duration of the LTSA spells, but also for the LTSA diagnosis, as it can affect service use frequency [19,22]. The objective of this study is to follow three central service schemes for supporting occupational ability—outpatient health care, inpatient care and rehabilitation—before and during an LTSA spell at least 30 days long. In addition, this study compares the use of the three service types between groups defined by employment and disability pension (DP) transition after an LTSA, taking demographic, socioeconomic and chronic disability-related covariates into account.

## 2. Materials and Methods

### 2.1. Study Population

Register-based data were collected from several registers for the years 2014–2018 for the total population of the city of Oulu, situated in Northern Finland [36]. With a population of 209,197 inhabitants in 2021, Oulu is the fifth largest city of Finland. Oulu does not differ in any systematic way from Finland as a whole on various demographic, socioeconomic or health care-related indicators [36].

Data on residency, demographics, socioeconomic status and LTSA spells were retrieved from registers of the Social Insurance Institution of Finland (Kela, Helsinki, Finland). Residents of Oulu who were 18–58 years old, not on a pension or a student at the start of 2015, were first included in the data (*N* = 73,766). The age limits were set so that all the subjects would be of adult age and would not reach the lowest limit of old-age pension in Finland (63 years) during the follow-up. Those receiving a pension were excluded, as pensioners are not entitled to sickness allowance. Students were excluded as our data on outpatient health care lacked information on student health care. Then, persons who started an LTSA spell lasting at least 30 days during 2015 but had no previous LTSAs during 12 months prior to the spell were finally included in the study (*N* = 2427). A flowchart of the whole inclusion/exclusion process of study subjects is presented in Appendix A (Appendix A).

### 2.2. Long-Term Sickness Absence (LTSA) and Disability Pension

Long-term sickness absence (LTSA) was measured through compensated sickness allowance days. Kela can pay sickness allowance to non-retired persons aged 16–67 as compensation for loss of income due to sickness or impairment. The allowance can be paid, when the sickness absence exceeds 10 working days, covered by the employer. A physician’s sickness certificate is needed for the allowance. Based on a certain diagnosis, the allowance can generally be paid for one year during two years’ time. Register data on sickness allowance spells were derived from Kela, including the start and end dates and diagnoses of LTSA spells. LTSA spells ≥ 30 days were studied, as longer sickness absences both signal a need for care or rehabilitation and are more significant risks for permanent disability [5,37].

A disability pension may be considered after the statutory maximum period of LTSA. Data on permanent disability pensions 2015–2018 was derived from registers of Kela and the Finnish Center for Pensions, including the start dates of DP.

### 2.3. The Follow-Up Setting

The start of the first LTSA spell in 2015 was set as baseline. The study population was followed for four years in total: 4 three-month periods (one year in all) before and 12 three-month periods (3 years in all) after the start of the LTSA spell. The three-year follow-up from baseline was chosen, since the third year after the start of LTSA is often a period for gaining disability pension, or returning to work, if the LTSA spell reaches its maximum length (see above). The visit to obtain a sickness certificate from a physician (first day of illness), needed for the sickness allowance, was included in the first follow-up period. 

### 2.4. Grouping Based on Employment and Disability Pension Transition after an LTSA

The study population was divided into four groups based on employment and disability pension grants during the last 12 months of the follow-up. This third follow-up year is timed after an LTSA for all subjects, as two years has passed since the start of the LTSA, and sickness allowance can be received maximally over two years’ time (see above). Employment spells for 2015–2018 were retrieved from registers of the Finnish Centre for Pensions.

Group 1 (*N* = 189) transferred to disability pension (DP) after the ≥30-day LTSA and by the end of the follow-up. Group 2 (*N* = 1639) were mostly employed after an LTSA—at least half of the last 12 follow-up months. Group 3 (*N* = 169) had some employment, at least 30 calendar days, but less than half of the last 12 months. Group 4 (*N* = 430) were not employed after an LTSA. This group had fewer than 30 employment days during the last 12 follow-up months. These LTSA groups are also presented in Appendix A (Appendix A).

### 2.5. Data on Outpatient and Inpatient Health Care and Rehabilitation

Data on the use of outpatient health care was collected for the years 2014–2018 covering all schemes of the Finnish service system (public, occupational, private). Data on public health care use was provided by the municipality of Oulu and the Care Register for Health Care [38]. The data on public care included visits to municipal health centres and outpatient visits to hospital-based specialized care. Data on occupational health service (OHS) visits were gathered from the four largest OHS providers in Oulu (Terveystalo, Mehiläinen, Attendo and Työterveys Virta), estimated to cover around 92% of employee clients entitled to OHS in Oulu [39]. Finally, data on private outpatient care visits were retrieved from the reimbursement registers of Kela.

In Finland, public outpatient primary health care is offered for all residents of municipalities in health centres. For the working population, however, OHSs are the main provider of primary care; all employees are entitled to preventive care, provided by the employer, and employers frequently also provide primary care through OHS [40]. Private health care is state-supported via partial reimbursement. The reimbursement varies, but is around one seventh for a general practitioner consultation. The role of the private scheme is growing but still rather small, due to the strong and affordable public and OHS schemes. Both public and private schemes, and to a small extent OHS, provide outpatient specialized care.

Face-to-face visits, phone calls and virtual consultations were included as they are active visits to health care professionals. Dental care and laboratory visits were excluded to harmonize the data between the service schemes. The number of outpatient visits was approximated by counting separate attendance days with each provider, as separate visits with the same date were inconsistent in the different register holders’ data. Then, the total number of attendance days was calculated for each subject, and for each three-month period.

Data on inpatient care were obtained from the Care Register for Health Care. The inpatient care periods included both hospitalization and inpatient care in public health centres. The total number of days in inpatient care for each three-month period was calculated for each subject.

Rehabilitation periods were studied using data of rehabilitation benefit spells from the registers of Kela and the Finnish Centre for Pensions. The benefit’s intended use is to secure income during vocational or medical rehabilitation. The total number of days in rehabilitation for each three-month period was calculated for each subject.

### 2.6. Covariates

Sex, age, marital status, socioeconomic status and entitlement to reimbursements for medicine expenses in 2014 were retrieved from registers of Kela. Unemployment benefit spells were retrieved for 2014 and 2015 from registers of Finnish Centre for Pensions. 

The study population was classified into four age groups (see Table 1). Marital status was categorized as married, unmarried, and divorced, separated or widowed. Socioeconomic status was measured in terms of occupational class. Occupational class distinguished between upper and lower non-manual employees, manual workers, entrepreneurs, and others following the classification of Statistics Finland [41]. The occupational class “others” included the long-term unemployed and persons without a statistical classification. Labour market status at the start of the LTSA was defined as employed, unemployed (on unemployment benefit) or other. 

Entitlement to reimbursements for medicine expenses was used as a proxy measure for chronic or severe disease [42]. These entitlements are ensured through National Health Insurance and guarantee the recipients’ access to medicines needed for the treatment of certain long-term diseases at a reasonable cost. A division between no diseases, one disease, and multiple chronic diseases (entitlements) was used. The total length of LTSA during the follow-up was counted and classified into ‘under two months’, ‘two months to eleven months’, and ‘maximum length. The study population was classified according to the diagnosis group of their first LTSA spell. This was carried out according to the International Statistical Classification of Diseases and Related Health Problems [43]. Diagnosis groups were mental disorders (‘mental LTSA’), musculoskeletal diseases (‘musculoskeletal LTSA’), and other diagnoses for LTSA.

### 2.7. Statistical Methods

The average, unadjusted number of outpatient health care visits, number of days in inpatient care, and the number of days in rehabilitation for each of the 4 three-month periods before and for each of the 12 three-month periods after the start of LTSA were first calculated. Covariate-adjusted estimates for the use of these three services for each period were then calculated using negative binomial regression models. This method is suitable for count data with a right-skewed distribution [44]. Finally, the association of the LTSA groups and covariates with the level of use for each service type during the three years after the start of the LTSA spell were analysed with negative binomial regression models. For these models, incidence rate ratios (IRRs) and predicted means with their 95 % confidence intervals are presented. The analyses were conducted using Stata statistical software package version 14.1 [45].

## 3. Results

### 3.1. Characteristics of the LTSA Groups

There were clear differences in the distributions of the covariates between the four LTSA groups (Table 1). Compared to others, persons in group 1 (transfer to DP after an LTSA) were on average older, had more chronic or severe diseases, and more often reached the statutory maximum period of LTSA during the follow-up. Persons in group 2 (mostly employed after an LTSA) were more often 31–50 years old, married, non-manual employees, and employed at the start of LTSA, than other groups. Persons in group 3 (some employment after an LTSA) were more often 18–30 years old than other groups, often unmarried, and had quite a similar socioeconomic profile to group 1. Persons in groups 2 and 3 were more often females and had a shorter total amount of LTSA than persons in other groups. Finally, those in group 4 (not employed after an LTSA) were often unmarried, unemployed at the start of LTSA, and had an LTSA based on a mental disorder more often than other groups.

### 3.2. Unadjusted Averages for Outpatient Health Care, Inpatient Care and Rehabilitation Use

Figure 1 presents the unadjusted averages for the number of outpatient health care visits, days in inpatient care and days in rehabilitation during each three-month period of the follow-up. Outpatient health care visits (Figure 1a) started to increase 4 to 6 months before the start of the LTSA spell that was set as baseline. The number of visits peaked when the LTSA spell started. That peak was slightly higher for group 1 (mean 9.4 visits in three months) than others. After that the level of outpatient health care use decreased gradually for groups 1 and 4, and rapidly for groups 2 and 3. Group differences in the level of use remained stable over the three years after the start of the LTSA spell. Group 1 had the most outpatient care visits on average, whereas groups 2 and 3 had the least.

For average inpatient care days (Figure 1b), there was a similar peak in the three-month period beginning from the start of the LTSA spell, and the number of days in that period was highest in group 1 (mean 6.6 days). For all LTSA groups, the days in inpatient care then decreased rapidly, but for group 1 the level remained higher compared to others until around 1.5 years (months 16 to 18) after the start of LTSA. 

For rehabilitation days (Figure 1c), the group trajectories were less consistent over time, with variation between the time periods. Before LTSA and well into the LTSA spell there were no clear LTSA group differences. A year after the start of the LTSA spell (months 13 to 15) the average rehabilitation days started increasing for group 4, and continued to increase almost to the end of the follow-up. For group 1, average rehabilitation days decreased 12 months after the start of LTSA on, and were on average near to zero around months 22–24. For group 2, the amount of rehabilitation days was relatively low and stable. Group 3 showed a somewhat similar but lower peak after an LTSA than group 4. 

Adjusting for covariates between LTSA groups in each time period narrowed the LTSA group differences for all three service types (predicted means in Figure 2), and the differences proved mostly statistically non-significant (see confidence intervals). The early peak in the outpatient care use proved the same in size between the LTSA groups. The average level of outpatient health care use (Figure 2a) was higher for group 1 than group 2 in months 7 to 33. After adjusting for covariates, the visibly higher level of rehabilitation use for group 4 proved statistically non-significant (Figure 2c), with broad confidence intervals indicating large variation within group 4. Finally, the number of rehabilitation days was higher for group 2 than group 1 from month 22 onwards, reflecting the near-zero average among those transferring to DP.

### 3.3. The Association of LTSA Groups with Service Use after the Start of the LTSA Spell

The association of LTSA groups and covariates with the average number of outpatient health care visits, days in inpatient care and days in rehabilitation were examined for the whole three-year period after the start of LTSA. This was carried out since Figure 1 and Figure 2 showed LTSA group differences only for the time following the start of LTSA. Predictor variables were entered to the negative binomial regression models in three blocks. Model 1 included LTSA groups with the sex and age group as covariates. Model 2 added marital status, occupational class, and labour market status at the start of LTSA as covariates. Fully adjusted model 3 further added the number of chronic or severe diseases, LTSA length and LTSA diagnosis group as covariates. Predicted means with their 95% confidence intervals for the LTSA groups are presented in Table 2. Comprehensive results for the estimates of both LTSA groups and covariates (including incident rate ratios (IRRs)) are presented in Appendix A (Appendix A). In all nine models, the negative binomial models fitted the outcome distribution better compared to Poisson regression models (dispersion parameter alpha not equal to zero).

For outpatient health care visits, model 1, adjusted for sex and age, showed on average more visits for group 1, compared to other LTSA groups (predicted mean 7.6 visits). Group 4 also had more visits on average than groups 2 and 3. Adding marital status and the two socioeconomic covariates (model 2) did not change these statistically significant LTSA group differences. However, adding the disease-related covariates in model 3 changed the LTSA group differences: the level of health care use was higher for group 1 than groups 2 and 3, but no longer higher than group 4.

In model 1 for inpatient care in days, group 1 had, on average, more inpatient care days (predicted mean 2.0 days) than other LTSA groups, while group 2 had fewer days (predicted mean 0.3 days) than other groups. In model 2, the difference between groups 2 and 3 proved statistically non-significant. The fully adjusted model 3 further narrowed the LTSA group differences considerably. In model 3, only group 1 with more inpatient care days differed from other groups in a statistically significant way—other groups did not differ from each other.

Finally, in model 1 for the rehabilitation days, group 4 had, on average, more rehabilitation days (predicted mean 4.4 days) than group 2. In model 2, group 4 differed from both groups 1 and 2 with a higher amount of rehabilitation days. These LTSA group differences remained in the fully adjusted model 3.

The effects of covariates on the use of the three services varied (see Appendix A). The number of chronic or severe diseases and LTSA length were strongly associated with the use of all three services, and especially with the number of rehabilitation days. Mental LTSA was associated with most outpatient care visits. In fully adjusted models, the youngest age group had the most inpatient care and rehabilitation days (see IRRs). Furthermore, in fully adjusted models, entrepreneurs had fewer outpatient care visits than others. Upper non-manual employees had fewer outpatient health care visits than lower non-manual employees, and fewer rehabilitation days than manual employees.

## 4. Discussion

The aim of this study was to understand how the use of outpatient and inpatient health care and rehabilitation develop one year before and three years after the start of long-term sickness absence (LTSA), and how the use of the three service types depends on the labour market situation after the LTSA in terms of disability pension and employment. We utilized extensive register data for non-pensioned, working age residents of Oulu, a city in Finland, with a ≥30-day LTSA spell in 2015 (*N* = 2427).

### 4.1. Outpatient and Inpatient Health Care

Outpatient health care visits and inpatient care days showed a relatively similar temporal pattern. Regardless of the labour market outcome of the LTSA, the level of service use peaked when the LTSA started. After that, the use gradually decreased until the end of the follow-up.

Persons who were mostly, or to some extent, employed during the year after an LTSA had approximately the same number of outpatient care visits as future disability pensioners and persons not returning to employment at the start of the LTSA, but had a low outpatient use level from that point onward. This early peak in health care use may be due to their access to occupational health services (OHS), since over 90% of this group was employed already at the start of the LTSA. It is widely recognized that those with access to OHS in Finland receive better and faster care [46]. In addition, a better socioeconomic position removes financial barriers, and enables access to quality care (e.g., private care) [47,48,49]. Interestingly, adjusting for covariates, this early peak remained the same relative to other LTSA groups, indicating that the peak may reflect sufficient early care aiding the return to work. The rapidly decreasing and low average outpatient care use after that peak reflects fast recovery and short LTSA spells in these groups, most likely related to milder health conditions and working conditions that enable early return to work. In addition, a covariate-adjusted analysis showed that persons employed after an LTSA used outpatient health care on average less frequently during the three-year period after the start of LTSA compared to persons transferring to DP.

Unadjusted results showed that persons who transferred to DP after an LTSA had the highest peak level at the start of LTSA in both outpatient and inpatient health care use. Additionally, when examining the whole three-year period from the start of LTSA and adjusting for covariates, persons who transferred to DP showed higher average levels of outpatient and inpatient health care use than the non-retiring LTSA groups. If outpatient and inpatient health care signal ill health, these results indicate that persons with disabling health problems are successfully identified by health care and pension systems. Studies on health symptom trajectories [50] and the psychotropic drug consumption [51,52] of retirees have shown a steep rise in the disability indicators before the pension grant, and a steady long decline after the pension grant. In this study, however, the decline started soon after the start of LTSA. For outpatient health care, this is partly due to the fact that certification from a physician is needed for the sickness allowance, and this is demonstrated by a peak in the number of visits. More generally, a higher level of health care use for future DP retirees may reflect a good standard of care. Those who are to transfer to DP possibly benefit from a more thorough attempt to improve their functional capacity during LTSA. A greater number of outpatient health care visits may also mean more accurate documentation of occupational disability, increasing chances for a disability pension award. These interpretations are not mutually exclusive and can thus all play a role in the results.

Adjusting for covariates mostly removed the LTSA group differences for the three-month interval measures and narrowed LTSA group differences when examining average service use during the three years after the start of LTSA. This shows that much of the differences in LTSA group levels were due to differences in demographic, socioeconomic, or disability-related background. For instance, those transferring to DP were older than other LTSA groups, and older age was associated with more frequent health care use in general [7,19]. Similarly, the differences between persons employed and persons not employed after an LTSA were narrower in the adjusted results, since a lower socioeconomic status is associated with very frequent health care use in general [15,16,17].

### 4.2. Rehabilitation

For rehabilitation use, the trajectories were different from those concerning health care. The trajectories were also quite different between the LTSA groups as there were no clear simultaneous peaks in the service use. Instead, for individuals not employed after an LTSA, the average rehabilitation days started increasing approximately one year after the start of LTSA. The increase continued almost to the end of the follow-up, i.e., after the LTSA. Half of this group was unemployed already at the start of their LTSA. The unemployed do not have access to occupational health services who specialize in occupational ability. It is known that slower care, ref. [46] less frequent care and poorer documentation of the condition due to lack of access to OHS may lead to both over-representation of unemployed persons among the DP applicants but under-representation among those who are granted the pension [53]. Further, there is a greater risk of marginalization among those unemployed if work ability is not regained after an LTSA [54]. However, our results indicate that the lack of access to quality care does not mean lesser access to rehabilitation. The unemployed do receive rehabilitation relatively often, but rather late in the process. A risk for prolonged and late rehabilitation among persons with a lower socioeconomic position has also been shown by Madsen [32]. The higher frequency of rehabilitation naturally signals not only access to, but the need for, rehabilitation. Individuals with a lower educational level or socioeconomic status tend to have poorer health [55,56] and are over-presented in rehabilitation activities in general [31,32]. Here, persons not employed after an LTSA had more than the average number of rehabilitation days after the start of LTSA even when all covariates, including chronic morbidity, were adjusted for. Persons with some employment after an LTSA showed a somewhat similar but a weaker peak in rehabilitation use after an LTSA, presumably for the same reasons as persons not employed. 

Among individuals who later transferred to DP—even if this group had the most health care, and supposedly had the most limiting medical conditions—there were, interestingly, only relatively few rehabilitation days during the year prior to an LTSA or the 12 months after the start of LTSA. Possibly a part of the conditions causing DP are very difficult to rehabilitate, and this is understood by pension insurers and other rehabilitation actors already when the rehabilitation chances are evaluated during an LTSA. Those with fewer work-years ahead of them and a medical condition requiring more effort may have less motivation for rehabilitation as well, reducing rehabilitative actions [57]. On the other hand, directing clients to at least occupational rehabilitation may be insufficient, as register- and document-based studies on disability pensioners in Finland have shown [26,27]. In Finland, those in rehabilitation have often experienced that the rehabilitation is provided too late, and this experience has been shown to be associated with a lower probability of returning to employment [58]. In our study, rehabilitation days decreased further during the follow-up, as this group started to transfer to DP. 

Individuals mostly employed after an LTSA had a steady low level of rehabilitation, a result echoing those concerning health care use. For those returning to work, the disabling medical conditions are deemed to be less severe and chronic, requiring care or rehabilitation less frequently compared to other individuals on LTSA. 

### 4.3. Strengths and Limitations

Our study population was based on register data of the total working-age population of the city of Oulu, Finland. We were able to utilize register data on date-level health care and rehabilitation use, LTSA spells, disability pensioning, and covariates. Especially, the rarely used register data on health care, based on comprehensive registers and covering all schemes relevant to the Finnish working-age population, strengthens the validity of our study. Registers are deemed to be highly reliable and objective, with no self-report bias and no loss to follow-up.

However, a limitation is the restriction of our study population to residents of one city, and to individuals with no LTSA or pensions in the previous year. This of course warrants caution in generalizing the results to the whole working-age population that experience LTSA in Finland or to other countries. Internationally, the results are probably best generalizable to countries with roughly similar benefit and health care systems, for example, the Nordic countries. Similar international studies are needed to show whether the findings are generalizable to other systems and contexts. A second limitation is the setting, not enabling us to gain insight into the actual consequences of the three service types, let alone the causality between the services and the outcome of the LTSA spell. A further limitation is the criteria behind our LTSA grouping. We opted for dividing non-retiring subjects based on the primary employment status during the last follow-up year, rather than actual labour market transitions. This was due to the modest size of the study population. Finally, considering the group not employed after an LTSA, the follow-up years were clearly insufficient to show the role of rehabilitation as the rehabilitation frequency peaked in the last 1.5 years for this group. Therefore, we could not detect the possible labour market consequences of the rehabilitation for this group. As positive labour market transitions after sickness absence may often require time, longer follow-up settings in general would benefit future studies. Similarly, longer timelines concerning pre-LTSA health care or rehabilitation use could be beneficial in identifying early risk factors for disability.

### 4.4. Practical Implications

Our results indicate that those not employed after an LTSA receive rehabilitation relatively often, but rather late in the disability process. Providing those outside employment preventive outpatient care—that is, equally quickly and with a clinician who has equal expertise on occupational ability issues as in OHS—may decrease the need for rehabilitation for some clients and advance direction to early rehabilitation for others. 

Legislative and administrative reforms could aid the group-specific challenges implied by our study. In addition to providing equal care for the unemployed, such improvements could be bringing down the waiting times in non-urgent public care, and directing those with lowered occupational ability to rehabilitation measures earlier in the process. In Finland, the government has made a proposal for care guarantee legislation for bringing down the maximum waiting times in non-urgent public care [59], possibly increasing preventive care for socioeconomic groups mainly using public health services. Those transferring to disability pension after an LTSA had relatively low frequency of rehabilitation even if health care use was frequent. It is possible that part of the pensions could be avoided by better direction to rehabilitation, as indicated by earlier studies [26,27,58]. 

## 5. Conclusions

Studying outpatient and inpatient health care and rehabilitation use one year before and three years after the start of long-term sickness absence, there were group difference based on the post-LTSA labour market situation: those who returned to employment after an LTSA seemed to have been provided with instant outpatient care for disability, aiding their return. For those not returning to employment after an LTSA, rehabilitation was used rather late in the disability process. For future disability pensioners, on the other hand, the high use of outpatient and inpatient health care was consistent with their increasing health problems leading to retirement. However, those who ended up on a disability pension were relatively rarely in rehabilitation before the pension. Legislative and administrative reforms could aid the group-specific challenges implied by our study. Such improvements could provide equal care for the unemployed, reducing waiting times in non-urgent public care and directing those with lowered occupational ability to rehabilitation measures earlier in the process.

## Figures and Tables

**Figure 1 ijerph-19-04990-f001:**
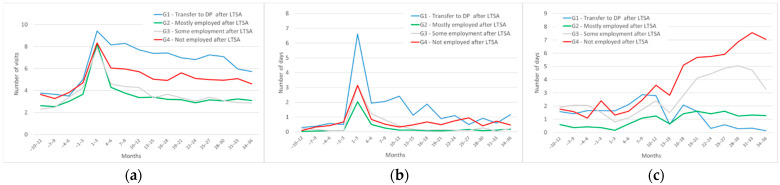
The use of three service types in the four LTSA groups. (**a**) Number of outpatient health care visits; (**b**) number of days in inpatient care; (**c**) number of days in rehabilitation.

**Figure 2 ijerph-19-04990-f002:**
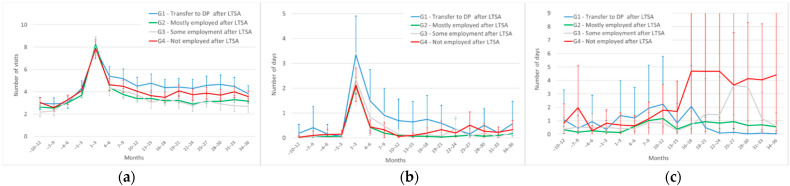
Covariate-adjusted * use of three service types in the four LTSA groups (predicted means and 95% confidence intervals). (**a**) Number of outpatient health care visits; (**b**) number of days in inpatient care; (**c**) number of days in rehabilitation. Note: * Sex, age group, marital status, occupational class, labour market status at the start of LTSA, chronic or severe diseases, LTSA length, and LTSA diagnosis group.

**Table 1 ijerph-19-04990-t001:** The covariates in the study population and by LTSA group.

	All	Group 1—Transfer to DP after LTSA	Group 2—Mostly Employed after LTSA	Group 3—Some Employment after LTSA	Group 4—Not Employed after LTSA
	*N* = 2427	*N* = 189	*N* = 1639	*N* = 169	*N* = 430
	%	%	%	%	%
Sex					
Male	42.3	50.8	40.4	43.8	45.1
Female	57.7	49.2	59.6	56.2	54.9
Age group					
18–30	20.2	2.1	16.8	39.6	33.3
31–40	24.2	4.2	27.0	21.9	23.3
41–50	27.0	17.5	30.1	17.8	23.3
51–58	28.6	76.2	26.1	20.7	20.2
Marital status					
Married	46.2	44.4	51.9	37.9	28.8
Unmarried	36.6	28.6	32.5	49.7	50.9
Divorced/separated/ widowed	17.1	27.0	15.7	12.4	20.2
Occupational class					
Upper non-manual employee	15.6	7.9	19.8	10.7	5.1
Lower non-manual employee	34.2	23.3	41.5	28.4	13.7
Manual worker	22.9	24.3	24.8	24.3	14.7
Entrepreneur	5.3	5.8	5.1	4.7	6.1
Other	21.9	38.6	8.9	32.0	60.5
Labour market status at the start of LTSA					
Employed	77.3	61.4	91.7	66.3	33.3
Unemployed	17.2	33.9	6.4	21.3	49.3
Other	5.6	4.8	17.9	12.4	17.4
Chronic or severe diseases					
No diseases	72.4	44.4	75.4	75.7	72.1
One disease	19.3	33.9	17.9	18.9	18.4
Multiple diseases	8.3	21.7	6.8	5.3	9.5
LTSA length					
Under 2 months	31.1	2.1	38.6	31.4	15.1
Two months to 11 months	48.5	28.6	53.8	53.3	34.9
Maximum length (one year)	20.4	69.3	7.6	15.4	50.0
LTSA diagnosis group					
Mental LTSA	20.8	14.3	15.8	26.6	40.2
Musculoskeletal LTSA	26.6	32.3	27.7	23.1	21.2
Other diagnosis LTSA	52.7	54.4	56.5	50.3	38.6
All	100.0	100.0	100.0	100.0	100.0

**Table 2 ijerph-19-04990-t002:** Negative binomial regression analysis models. The LTSA groups’ expected number of outpatient health care visits, days in inpatient care and days in rehabilitation after the start of LTSA in (predicted means and 95% confidence intervals).

	M1		M2		M3	
	Predicted Means	95% CI	Predicted Means	95% CI	Predicted Means	95% CI
The Expected Number of Outpatient Health Care Visits						
G1—Transfer to DP after an LTSA	7.56	6.75–8.36	7.43	6.62–8.23	4.87	4.36–5.38
G2—Mostly employed after an LTSA	3.65	3.51–3.79	3.66	3.51–3.81	3.79	3.65–3.94
G3—Some employment after an LTSA	3.96	3.49–4.43	3.94	3.47–4.40	3.69	3.28–4.10
G4—Not employed after an LTSA	5.57	5.18–5.97	5.51	5.06–5.96	4.22	3.89–4.55
The Expected Number of Days in Inpatient Care						
G1—Transfer to DP after an LTSA	1.95	1.47–2.43	1.79	1.34–2.23	0.84	0.62–1.06
G2—Mostly employed after an LTSA	0.31	0.28–0.35	0.32	0.28–0.36	0.29	0.26–0.33
G3—Some employment after an LTSA	0.52	0.36–0.68	0.49	0.34–0.63	0.40	0.28–0.52
G4—Not employed after an LTSA	0.77	0.63–0.90	0.62	0.50–0.75	0.42	0.33–0.51
The Expected Number of Days in Rehabilitation						
G1—Transfer to DP after an LTSA	1.59	0.46–2.73	1.08	0.30–1.87	0.40	0.12–0.68
G2—Mostly employed after an LTSA	1.11	0.86–1.36	1.06	0.82–1.30	0.76	0.60–0.93
G3—Some employment after an LTSA	2.72	0.82–4.63	2.41	0.65–4.17	1.29	0.37–2.21
G4—Not employed after an LTSA	4.42	2.42–6.40	4.08	2.21–5.59	2.22	1.28–3.16

M1: Adjusted for sex, age group. M2: Adjusted for sex, age group, marital status, occupational class, labour market status at the start of LTSA. M3: Adjusted for sex, age group, marital status, occupational class, labour market status at the start of LTSA, chronic diseases, LTSA length, LTSA diagnosis group (fully adjusted model).

## Data Availability

Data cannot be shared publicly because strict restrictions apply to the availability of confidential individual-level register data. These analyses were conducted with permissions from third-party data holders for the current study. Permissions to obtain register data from the City of Oulu, from the Social Insurance Institution of Finland (Kela), from Finnish Centre for Pensions and from the occupational health care providers may be applied for scientific research purposes from the Finnish Health and Social Data Permit Authority Findata (https://www.findata.fi/en/, accessed on 1 March 2022). A license to obtain register data from Statistics Finland may be applied for separately (https://www.tilastokeskus.fi/meta/tietosuoja/kayttolupa_en.html, accessed on 1 March 2022).

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
