# Peer review of "Use of Health Services and Rehabilitation before and after the Beginning of Long-Term Sickness Absence—Comparing the Use by Employment and Disability Pension Transition after the Sickness Absence in Finland"

_ijerph, 2022, doi:10.3390/ijerph19094990_

Round 1
Reviewer 1 Report
Thank you for the possibility to read this interesting manuscript. The authors have detected a relevant topic and addressed the research questions with solid methodology. However, some concerns remain that the authors could consider addressing.
Abstract
Please provide some estimates to support your results section.
Introduction
Please define health care use and rehabilitation services. Please note that you have also used the term rehabilitation actions, early vocational or medical rehabilitation, rehabilitation services, and rehabilitation. Please harmonize and systemize.
Introduction
Finland is highlighted which might be relevant since the study utilizes data from one city in Finland. However, readers of IJERPH would benefit from more broad perspective. What about Nordic countries or other world? Please see e.g., systematic literature review by Andreas Norlund et al. and other similar papers about rehabilitation and sick leave.
Material and methods:
2.2. LTSA and disability pension
Could you reconsider the structure of this material and methods section? LTSA is described both in the 2.1. Study population and 2.2. LTSA and DP paragraphs which makes the text and descriptions very hard to follow. Was DP included with LTSA (i.e. extremely long LTSA?) or was it a separate measure? Please clarify.
2.2. LTSA and DP
Do you have data for diagnosis? Could that be included? Sorry, that information comes in the covariates? Please see my earlier comment on restructuring the material and methods section for clarity.
2.3. The follow-up setting
Why you use unbalanced before-after design? Why only one year before LTSA and 3 years after? Please address the effect on this unbalanced design to your results in the discussion section. Could you add a sensitivity test for other baseline years than 2015?
2.4. Grouping based on employment and disability pension transition after LTSA
The grouping has been described in text, but no table is provided? Please consider adding a table (perhaps supplemental material) to provide more details for the grouping as this is very essential for your study.
2.7. Statistical analyses
Why the time periods were divided into three month periods? You have decided to use pre-selected groups. Why not using group-based trajectory models instead?
Discussion
Please add at the beginning of discussion that you utilized data of one city in Finland (n = xxxx) for years 2014-2018. This would give the readers the context.
The authors state that “interestingly, the trajectories between persons mostly or somewhat employed after LTSA did not differ in their outpatient health acre use trajectories. Why this is interesting and why it should differ? I think the main “problem” here is the “slump category of outpatient care”. It seems to include everything, i.e., from virtual contacts to active visits. Could you perhaps add details to outpatient care? What about specialized outpatient care? Or comparison between care providers? Although it would be interesting – as the authors say – to investigate outpatient care, I think the measure itself is rather complex and may achieve results without meaning.
Please add discussion on your unbalanced time periods before and after LTSA. Furthermore, please consider the selection of 3 years of follow-up and the use of three month periods for measures.
The discussion lacks consideration of generalizability. How these results based on one city in Finland can apply to other cities or parts of Finland? What about generalizability to other countries?
Conclusion
I would advice not to address legislative reforms in the conclusions, hence the two last sentences could be rather moved to the discussion section.
Author Response
We thank the referee for the valuable and well-informed comments.
Abstract
Please provide some estimates to support your results section.
We have now included the predicted means for group 1 in the Abstract, as this is the main finding.
Introduction
Please define health care use and rehabilitation services. Please note that you have also used the term rehabilitation actions, early vocational or medical rehabilitation, rehabilitation services, and rehabilitation. Please harmonize and systemize.
We have now deleted expressions “rehabilitation action” and “rehabilitation service” from the introduction to be more consistent. Our perspective to the rehabilitation is quite broad and unspecified, as the measure we use – rehabilitation benefit – can cover a wide range of rehabilitation programmes. This is expressed in the “Materials and Methods”. In Introduction, we are accordingly forced to stay on the general level, but draw earlier findings from studies concerning vocational, medical, and multidisciplinary rehabilitation interventions. We hope this is appropriate and acceptable for publishing.
Introduction
Finland is highlighted which might be relevant since the study utilizes data from one city in Finland. However, readers of IJERPH would benefit from more broad perspective. What about Nordic countries or other world? Please see e.g., systematic literature review by Andreas Norlund et al. and other similar papers about rehabilitation and sick leave.
Thank you for this suggestion. The paragraph in question motivates our temporal examination of the rehabilitation use in relation to the disability spell. As the studies on timing of rehabilitation that we could find were Finnish, they were included in the paragraph. However, we have now added two international references on the effectiveness and timing of rehabilitation to this paragraph (including Nordlund et al., as suggested), and hope it now captures a broader perspective.
Material and methods:
2.2. LTSA and disability pension
Could you reconsider the structure of this material and methods section? LTSA is described both in the 2.1. Study population and 2.2. LTSA and DP paragraphs which makes the text and descriptions very hard to follow. Was DP included with LTSA (i.e. extremely long LTSA?) or was it a separate measure? Please clarify.
We have restructured the chapter. Now LTSA is mainly described only in the LTSA and disability pension –subchapter. Information on the disability pension data used for the LTSA grouping is now described in more detail (data included start dates for the disability pensions). We hope the chapter is now more understandable. We have also added a flowchart on the definition of the study population as a Supplement, to increase clarity.
2.2. LTSA and DP
Do you have data for diagnosis? Could that be included? Sorry, that information comes in the covariates? Please see my earlier comment on restructuring the material and methods section for clarity.
We used the diagnosis of the first LTSA spell in 2015 to classify the study population into three LTSA diagnosis groups. This is hopefully now understandable in the text, as the information on diagnosis is mentioned already in the LTSA and disability pension subchapter. The diagnosis-based grouping is described in the covariates subchapter, since it was used only as a covariate.
2.3. The follow-up setting
Why you use unbalanced before-after design? Why only one year before LTSA and 3 years after? Please address the effect on this unbalanced design to your results in the discussion section.
The choice to divide the follow-up time into one year before and three years after the start of the LTSA spell was guided by the nature of the LTSA and disability process, as explained below.
The three-year follow-up after the start of the LTSA spell is used so that labour market status/ disability pension after LTSA can be examined in the Finnish context. Three years capture the time period when a) the maximum LTSA period is reached by latest [“the allowance can generally be paid for one year during two years’ time”] plus b) those who as a consequence of the maximum LTSA time apply and transit into disability pension. Because of this process, the third follow-up year was pivotal for counting the amount of post-LTSA employment.
Concerning the one-year pre-LTSA follow-up: Our main focus was in the time starting from the beginning of the LTSA spell, reflected in the way the Introduction is built. In addition, we included persons in the study who had no LTSA during 12 months prior to the 2015 LTSA spell. This was made to have a specific study population, those on a new or first LTSA spell. Thus it was also natural to examine the retrospective 12 months as the backwards time line.
In the manuscript, we discuss the sufficiency of a three-year follow-up from the start of LTSA, especially concerning the relation of rehabilitation and group 4 (being outside employment) in the end of Strengths and limitations chapter. We have now added a sentence on the more general sufficiency of the follow-up time after that sentence.
Could you add a sensitivity test for other baseline years than 2015?
Unfortunately, since the number of observation years in our data set is limited, our data does not enable a similar setting for other years.
2014-2018 are the years from which we have reliable data on covariates, LTSA spells, DP spells, health care and rehabilitation services. For example, since we do not have data for 2019, we cannot set the baseline LTSA year to be 2016 since a three-year-long follow-up time is needed after the start of the LTSA spell. Similarly, we cannot set 2014 as baseline as we do not have reliable data on 2013 OHS health care, making the examining of outpatient health care impossible. We hope that this study setting is approved without this kind of sensitivity analysis that we are unfortunately unable to provide.
Furthermore, as the Finnish health care, rehabilitation, sickness allowance or disability pension systems have not changed during focal years, it is very unlikely that the results concerning group-based service use differences or temporal development during LTSA would show different results with one or two year changes in the baseline.
2.4. Grouping based on employment and disability pension transition after LTSA
The grouping has been described in text, but no table is provided? Please consider adding a table (perhaps supplemental material) to provide more details for the grouping as this is very essential for your study.
We have now added a clarifying Supplement table describing the four groups, as suggested.
2.7. Statistical analyses
Why the time periods were divided into three month periods?
First, we have used three-month intervals, i.e. quarters of a year, as they are intuitive but also because 3- to 4-month intervals are quite common in research literature dealing with disability and return to work (see i.a. Laaksonen et al. 2012; Leinonen et al. 2013; Madsen 2020, below)
In addition, the three-month interval was considered frequent enough, and a clear way to follow the service use. With a more frequent follow-up interval (for instance 1 month) the visual presentation of results would be more difficult, especially with confidence intervals.
Laaksonen M, Metsä-Simola N, Martikainen P, Pietiläinen O, Rahkonen O, Gould R, Partonen T, Lahelma E. Trajectories of mental health before and after old-age and disability retirement: a register-based study on purchases of psychotropic drugs. Scandinavian journal of work, environment & health. 2012 Sep 1:409-17.
Leinonen T, Lahelma E, Martikainen P. Trajectories of antidepressant medication before and after retirement: the contribution of socio-demographic factors. European journal of epidemiology. 2013 May;28(5):417-26.
Madsen AÅ. Return to work after first incidence of long-term sickness absence: a 10-year prospective follow-up study identifying labour-market trajectories using sequence analysis. Scandinavian Journal of Public Health. 2020 Mar;48(2):134-43.
You have decided to use pre-selected groups. Why not using group-based trajectory models instead?
Our pre-selected groups are rather intuitive in the occupational disability literature and discourse, as return to full work, return to partial employment, being outside employment, or transfer to permanent disability are main interests for policy actors. Pre-selected groups with their own service use trajectories also bring forth the association of post-LTSA labour market status and health/ rehabilitation service use in a concrete way.
Using trajectory models would indeed have been an alternative. However, the method often produces rather predictable groups (steady low, steady high, increasing, decreasing), and therefore using trajectory models makes the covariates’ effects of the trajectories the most vital part of the study. In that case, post-LTSA employment and disability pension would be separate independent explanatory variables.
In our view, building trajectories based on service use and then explaining these trajectory groups with, i.a. sickness absence, employment, and future disability pension would not showcase the different groups in the way that would show clear results on health care use according to the labour market status after the start of the LTSA spell. Instead, we opted for a method that first brings out “real” service use trajectories for pre-selected groups, and then produces similar graphs adjusting for covariates. Last, groups and covariates effects on service use was presented (Tables 2 + S Tables 2-4)
Discussion
Please add at the beginning of discussion that you utilized data of one city in Finland (n = xxxx) for years 2014-2018. This would give the readers the context.
We have now added a sentence “We utilized extensive register data for non-pensioned, working age residents of Oulu, a city in Finland, with a ≥30 day LTSA spell in 2015 (N = 2,427)” to the first paragraph of Discussion.
The authors state that “interestingly, the trajectories between persons mostly or somewhat employed after LTSA did not differ in their outpatient health acre use trajectories. Why this is interesting and why it should differ?
We expected these groups to differ from each other, based on previous knowledge on the role of the labour market inclusion on health-related outcomes. One explanation for our result that the groups did not differ could be that these groups both have on average a clearly shorter total LTSA than other groups (see Table 1). Thus even unadjusted results (Fig 1a) show low outpatient health care use for groups 2 and 3 after the initial peak. However, as all non-retiring groups 2, 3 and 4 did not differ in their covariate-adjusted average outpatient health care use (Table 2), we decided to erase the sentence altogether from the manuscript. We have extended the length of the manuscript on the basis of other referee comments, and want to avoid making the manuscript too long.
I think the main “problem” here is the “slump category of outpatient care”. It seems to include everything, i.e., from virtual contacts to active visits.
We have decided to include all contacts that require action from the patient and happening in real-time, including phone calls and virtual meetings. With the revolutionary change in health care consultation methods in the past decade, and with a substantial part of the visits happening without a physically same location, it could be in our view rather biased to include only on the spot meetings in the study.
Could you perhaps add details to outpatient care? What about specialized outpatient care? Or comparison between care providers? Although it would be interesting – as the authors say – to investigate outpatient care, I think the measure itself is rather complex and may achieve results without meaning.
Unfortunately, we could not specify the amount of each consulting methods or provide the exact amount of primary care versus specialized care. There are two reasons for this: First, we had to opt in counting visiting days (see the precise amount of separate attendance days with each health care provider), which makes these specifications impossible. Second, the outpatient care visits were counted from using many register data sets, covering all three service sectors and four private OHS actors. As these data sets are not harmonious, and do not provide the info on specialized care with the same reliability, these figures could not be counted for the whole data.
Furthermore, considering the comparison between the providers/ service sectors: As we already had a quite complex study setting, comparing the sectors or providers of the services did not fit the present study.
Please add discussion on your unbalanced time periods before and after LTSA. Furthermore, please consider the selection of 3 years of follow-up and the use of three month periods for measures.
We have added brief consideration of the follow-up length to the follow-up setting subchapter (Material and Methods) and also to the end of Strengths and limitations subchapter in Discussion (see also referee’s comment about unbalanced follow-up setting above).
The discussion lacks consideration of generalizability. How these results based on one city in Finland can apply to other cities or parts of Finland? What about generalizability to other countries?
As the data were collected from only one city, the results may not be totally generalizable to
the total Finnish working-age population. Since Oulu is a large city, its population has more opportunities to use occupational health care and private sector health services than inhabitants of more sparsely populated areas. However, the health care system and rehabilitation systems are fundamentally similar in the whole country. Thus, the results are estimated to be generalizable to the whole of Finland and may provide insights to other countries with roughly similar systems.
We have considered the generalizability of the results to the whole Finland in Strengths and limitations section. We have now rephrased and extended that section a bit to consider the international generalizability better. To avoid extending the already long manuscript a lot, we have we have opted to keep this consideration short.
Conclusion
I would advice not to address legislative reforms in the conclusions, hence the two last sentences could be rather moved to the discussion section.
We have moved that text up to the Practical implications section.
Reviewer 2 Report
This paper purposed to follow three central service schemes for supporting occupational ability ─ outpatient health care, inpatient care and rehabilitation ─ before and during a long-term sickness absence (LTSA) spell at least 30 days long and to compare the use of the three service types between groups defined by employment and disability pension (DP) transition after LTSA, taking demographic, socioeconomic and chronic disability-related covariates into account. I do have some comments as listed below in the order noted.
Comment 1: The quality of the data set is very important, especially for the total population of the city of Oulu, situated in Northern Finland. For this reason, please clarify the inclusion criteria and exclusion criteria of sample collection in the Methods section and please provide a flowchart immediately at the subsection of Study Population.
Comment 2: Please provide the deviance/degrees-of-freedom statistic to assess the model fit in Table 2.
Comment 3: What is the novelty of this study although currently the similar article has been published in the BMJ Open 2022;12(2):e053948.
Author Response
We thank the referee for the valuable and well-informed comments.
Comment 1: The quality of the data set is very important, especially for the total population of the city of Oulu, situated in Northern Finland. For this reason, please clarify the inclusion criteria and exclusion criteria of sample collection in the Methods section and please provide a flowchart immediately at the subsection of Study Population.
Thank you for this suggestion. We have now produced a flowchart and placed it as Supplementary material. If needed, it can naturally be moved to the Study population section.
Comment 2: Please provide the deviance/degrees-of-freedom statistic to assess the model fit in Table 2.
The suggested deviance/Df statistics for Table 2 were as follows:
Outpatient health care
M1: Deviance (df=2418) 11770.3; M2: Deviance (df=2410) 11744.6; M3: Deviance (df=2404) 11204.0
Inpatient health care
M1: Deviance (df=2418) 4338.1; M2: Deviance (df=2410) 4298.1; M3: Deviance (df=2404) 4089.6
Rehabilitation
M1: Deviance (df=2418) 4298.6; M2: Deviance (df=2410) 4279.0; M3: Deviance (df=2404) 4167.5
Our main purpose was not to compare the model fits between Models 1, 2 and 3, but to examine if LTSA groups still have an effect on service use when all covariates are adjusted for. In our view, if a model fit comparison should be reported, we suggest the likelihood-ratio test of alpha (chi-square test). It compares the negative binomial regression solution to a regression run with Poisson model. More specifically, it tests if the dispersion parameter alpha is equal to zero. In the case of M3 for outpatient health care (LR test of alpha=0: chibar2(01) = 1347.03, Prob >= chibar2 = 0.000) the assumption that alpha is equal to zero is rejected (alpha is significantly greater than zero), than the data are over-dispersed and are better estimated using a negative binomial model than a poisson model. This also applied to all our models nine models (3 X 3) in Table 2. We have added this information to the Results as text.
For reasons stated above, and also for keeping Table 2 tidy, we suggest leaving dev/df information out of Table 2. If however, the deviances of the analyses are essential for referee, we of course will insert them to Table 2. A further Supplement table is also possible if required.
Comment 3: What is the novelty of this study although currently the similar article has been published in the BMJ Open 2022;12(2):e053948.
The article in BMJ Open certainly has common features with the present study. The data sample is the same. The research questions also do bear some similarities. However, the present study focuses on the labour market status/ amount of employment after LTSA as a main factor explaining the development and level of service use, whereas the previous study focused on the length of LTSA. Here that factor is used only as a covariate. This study also extends the focus on services significantly, including inpatient care and rehabilitation. All in all, we consider this an independent study with mostly different focus and conclusions.
Round 2
Reviewer 1 Report
Thank you for the revision, which has improved your manuscript. I would like to suggest the publication of this manuscript.